# *Anaplasma phagocytophilum* and Other *Anaplasma* spp. in Various Hosts in the Mnisi Community, Mpumalanga Province, South Africa

**DOI:** 10.3390/microorganisms8111812

**Published:** 2020-11-18

**Authors:** Agatha O. Kolo, Nicola E. Collins, Kelly A. Brayton, Mamohale Chaisi, Lucille Blumberg, John Frean, Cory A. Gall, Jeanette M. Wentzel, Samantha Wills-Berriman, Liesl De Boni, Jacqueline Weyer, Jennifer Rossouw, Marinda C. Oosthuizen

**Affiliations:** 1Vectors and Vector-Borne Diseases Research Programme, Department of Veterinary Tropical Diseases, University of Pretoria, Pretoria 0110, South Africa; nicola.collins@up.ac.za (N.E.C.); kbrayton@wsu.edu (K.A.B.); samwillsberriman@gmail.com (S.W.-B.); liesl.deboni@gmail.com (L.D.B.); marinda.oosthuizen@up.ac.za (M.C.O.); 2Department of Veterinary Microbiology and Pathology, Washington State University, Pullman, WA 99164, USA; 3Zoological Research, Foundational Biodiversity & Services, South African National Biodiversity Institute, Pretoria 0001, South Africa; m.chaisi@sanbi.org.za; 4National Institute for Communicable Diseases, Johannesburg 2192, South Africa; lucilleb@nicd.ac.za (L.B.); johnf@nicd.ac.za (J.F.); jacquelinew@nicd.ac.za (J.W.); jennyr@nicd.ac.za (J.R.); 5Valley Medical Centre, Lewiston, ID 83501, USA; gall.cory@gmail.com; 6Hans Hoheisen Wildlife Research Station, Faculty of Veterinary Science, University of Pretoria, Pretoria 0110, South Africa; jeanette.wentzel@up.ac.za; 7Department of Medical Virology, University of Pretoria, Pretoria 0084, South Africa

**Keywords:** *Anaplasma phagocytophilum*, *Anaplasma* spp., various hosts, *Rhipicephalus sanguineus*, bacterial community, genetic variation, rural population

## Abstract

DNA samples from 74 patients with non-malarial acute febrile illness (AFI), 282 rodents, 100 cattle, 56 dogs and 160 *Rhipicephalus sanguineus* ticks were screened for the presence of *Anaplasma phagocytophilum* DNA using a quantitative PCR (qPCR) assay targeting the *msp2* gene. The test detected both *A. phagocytophilum* and *Anaplasma* sp. SA/ZAM dog DNA. Microbiome sequencing confirmed the presence of low levels of *A. phagocytophilum* DNA in the blood of rodents, dogs and cattle, while high levels of *A. platys* and *Anaplasma* sp. SA/ZAM dog were detected in dogs. Directed sequencing of the 16S rRNA and *gltA* genes in selected samples revealed the presence of *A. phagocytophilum* DNA in humans, dogs and rodents and highlighted its importance as a possible contributing cause of AFI in South Africa. A number of recently described *Anaplasma* species and *A. platys* were also detected in the study. Phylogenetic analyses grouped *Anaplasma* sp. SA/ZAM dog into a distinct clade, with sufficient divergence from other *Anaplasma* species to warrant classification as a separate species. Until appropriate type-material can be deposited and the species is formally described, we will refer to this novel organism as *Anaplasma* sp. SA dog.

## 1. Introduction

*Anaplasma phagocytophilum* is a zoonotic tick-borne intracellular pathogen that causes granulocytic anaplasmosis in humans, dogs and horses, and tick-borne fever in ruminants [1]. Clinical signs of the disease in humans range from mild febrile illness to a life-threatening condition [2,3]. The main vectors are ixodid ticks, namely *Ixodes ricinus* in Europe, *I. scapularis*, and *I. pacificus* in the eastern and western parts of the United States, and *I. persulcatus* in Asia [4]. In Europe, *A. phagocytophilum* has been detected in the yellow-necked mouse [5] and field voles [6], while in the Eastern United States, the white-footed mouse is considered the main reservoir host [7].

There have been reports of the detection of *A. phagocytophilum* in Africa, mainly identified using nucleic acid-based detection methods. *Anaplasma phagocytophilum* DNA has been detected in horses, ticks and cattle from Tunisia [8,9], vervet monkeys and baboons in Zambia [10], lions, African wild cats and servals in Zimbabwe [11] and dogs and cattle in Algeria [12,13]. *Anaplasma phagocytophilum* DNA has also been detected in several tick species including *Rhipicephalus sanguineus* (Egypt) [14], *Amblyomma cohaerens* and *Rhipicephalus pulchellus* (Ethiopia) [15,16] and *Rhipicephalus maculatum* (Kenya) [17]. Although one study reported the detection of *A. phagocytophilum* in ticks collected from cattle, sheep and goats in South Africa [18], this finding cannot be considered valid because the primers used in that study can amplify any *Anaplasma* species. The sequence data presented (two alignments of 15 and 20 nucleotides) show only that the detected sequence differs from all other *Anaplasma* species, including *A. phagocytophilum*, by one nucleotide in each of the primer regions, and thus, was likely introduced during amplification, i.e., none of the data is discriminatory to the species level and shows, at best, that the authors detected an *Anaplasma* species.

While the preceding studies suggest that *A. phagocytophilum* is present throughout Africa, this conclusion may not be valid, since it is possible that previously unrecognized *Anaplasma* spp. might have been detected in these studies. The fragment sizes of the 16S rRNA gene amplicons and/or sequences generated in these studies were short (below 490 nucleotides), and since there are few nucleotide differences in the 16S rRNA gene between closely related *Anaplasma* species, these sequences may not have spanned species-discriminatory regions.

With the advent of high throughput sequencing methodologies and the recent plethora of 16S rRNA gene surveys, numerous distinct *Anaplasma*-like 16S rRNA gene sequences have been deposited in public databases. The relationship of these agents to known pathogens, and their ability to serve as a source of cross-reaction in molecular testing, has not been assessed. Therefore, caution must be used when assigning species within the *Anaplasma* genus, as the 16S rRNA genes are highly similar, frequently with identities of >98% [19], and when an identity score of 97% is applied (as is used in many studies), misclassification will ensue. Similar caution should be applied in the design and application of tests to detect *Anaplasma* species.

One example of an organism that is very closely related to, but distinct from, *A. phagocytophilum*, is *Anaplasma* sp. SA dog strain, which was detected in dogs from South Africa [20,21]. A similar organism, designated *Anaplasma* sp. ZAM dog, was detected in Zambia [22].

As the importance of *A. phagocytophilum* and its role in febrile illness in South Africa is unknown, we must develop methods that are rapid, sensitive and specific for the detection of this pathogen. In the present study, we examined 672 samples and confirmed detection of both *A. phagocytophilum* and *Anaplasma* sp. ZAM dog DNA in South Africa. Furthermore, the occurrence and genetic diversity of *A. phagocytophilum* in various hosts in Mnisi, an agro-pastoral community at the wildlife–livestock–human interface in South Africa, were explored.

## 2. Materials and Methods

### 2.1. Ethics Approval

The study was approved by the Animal Ethics Committee of the Faculty of Veterinary Science (V105-15, 15/11/2015), and the Human Ethics Committees of the Faculty of Health Sciences of the University of Pretoria (152/2016, 26/05/2016) and the University of the Witwatersrand (M120667, 17/06/2014). Rodent trapping, tick collection and use of cattle and dog biobanked samples were approved by the Department of Agriculture, Forestry and Fisheries under Section 20 of the Animal Diseases Act of 1984 (reference numbers 12/11/1/7/5, 12/11/1/1, 12/11/1/1/6).

### 2.2. Study Area

Samples were collected in the Mnisi community located in the north-eastern corner of the Bushbuckridge municipality in Mpumalanga Province, South Africa (Figure 1). The geographic coordinates are (24.8205° S, 31.1710° E). The Mnisi community shares 75% of its boundary with adjacent wildlife areas. Livestock farming is the main agricultural activity and more cattle are kept than any other livestock species. Most cattle owners also own dogs, which accompany herders in the field and to the cattle dip tanks. Rodents are widespread and abundant in the community [23].

### 2.3. Collection of Blood Samples

Blood samples were collected from 282 rodents from three habitat areas, urban/peri-urban (Gottenburg and Hlalakahle), communal rangelands (Tlhavekisa) and a protected area (Manyeleti) (Figure 1), during three research field trips in July/August 2014, January 2015 and September 2015 (Table 1). Rodents were trapped using baited Sherman traps, identified using a field guide [24] and humanely euthanized using ISOFOR^®^ isoflurane (1-chloro-2, 2, 2-trifluro-ethyldifluoromethyl ether; Safeline Pharmaceuticals, South Africa) by transferring them from the Sherman traps into transparent zip lock bags that contained a cotton wool ball that had been dabbed in the ISOFOR. Blood was collected in EDTA tubes and on FTA cards immediately after the last heartbeat. A list of all the rodent species captured from the habitat areas is provided in Appendix A.

Blood samples from 56 domestic dogs and 100 cattle collected in EDTA tubes and stored in the biobank at the Hans Hoheisen Wildlife Research Station were utilized. Blood samples from dogs and cattle were collected during a Health and Demographic Surveillance System (HDSS) program that ran in the Mnisi community [25]. Blood samples from apparently healthy, domestic dogs from households in Hluvukani village between April and May 2012. Blood samples from cattle at dip tanks in Seville A, Seville B, Hlalakahle, Tlhavekisa and Gottenburg (Figure 1) between April and September 2013. Cattle in the area go to the dip tanks weekly for dipping to control the tick burden and for veterinary inspection for foot and mouth disease. Blood samples that had been collected previously from 74 patients with non-malarial acute febrile illness (AFI) from Gottenburg, Utha and Welverdiend community clinics were used [26]. These samples were made available to the project by the National Institute of Communicable Diseases (NICD), Johannesburg, South Africa. Patient blood samples were collected from adults that presented with a documented axillary temperature, or a history of fever within the last 72 h. Patients with AFI were assessed by the clinic staff, a routine malaria smear was done and, if negative, the patients were referred to the study nurse for enrolment into the study. If the patient agreed to participate, a questionnaire related to his/her contact with animals, presence at dip tanks and tick bites was then completed; and two blood tubes, one coagulated for acute serology and an EDTA tube for molecular tests, were taken [26].

### 2.4. Collection of Ticks

Adult male *R. sanguineus* ticks (160) were manually collected from dogs in Athol and Hluvukani (Table 1; Figure 1). Tick collection was biased towards male ticks because it is known that male ticks tend to move more frequently between hosts in the quest for mating possibilities [27]. Ticks were identified to the species level using relevant taxonomic keys [28], pooled in groups of 8 (1 pool = 8 adults) and kept for 72 h to ensure digestion of the host blood meal. Ticks were surface sterilized by vigorous shaking in 5% bleach solution, then 5% ethanol solution followed by a final rinse with double distilled water and whole ticks were manually disrupted with a Tissue Lyzer^®^ (Qiagen, Hilden, Germany).

### 2.5. DNA Extraction, Quantitative Real-Time PCR and Assay Specificity

DNA was extracted from all samples using the QIAamp DNA mini kit ^®^ (Qiagen) according to the manufacturer’s instructions. DNA concentration and quality were evaluated using the BioTek^®^ Powerwave XS2 microplate spectrophotometer (Davies Diagnostics, South Africa). Sample DNA was screened for *A. phagocytophilum* using a previously-reported qPCR assay [29]. Primers, ApMSP2 forward (5′-ATG GAA GGT AGT GTT GGT TAT GGT ATT-3′), ApMSP2 reverse (5′-TTG GTC TTG AAGCGC TCG TA-3′) and a TaqMan probe, ApMSP2p (FAM-5′-TGG TGC CAG GGTTGA GCT TGA GAT TG-3′-TAMRA) were used to amplify and detect a 77 bp fragment of the *msp2* gene. Reactions were performed in a final volume of 20 µL comprising 1 × Taqman^®^ Universal PCR Master Mix (Thermofisher Scientific, South Africa), 900 nM of each primer, 125 nM of the probe and 2.5 µL of template DNA. The reactions were run on a StepOnePlus™ Real-Time PCR System (Applied Biosystems, Foster City, CA, USA), using the cycling conditions reported previously [29] with the modification of UNG incubation at 50 °C for 2 min, prior to cycling. Positive DNA from the L610 [dog] strain, an *A. phagocytophilum* strain isolated from a dog in Germany [30], and negative (PCR grade water) controls were included with each run. The results were analyzed using the StepOnePlus software version 2.2. The analytical specificity of the qPCR assay was evaluated using DNA from *A. marginale* (cattle field sample previously confirmed to be infected with *A. marginale*), *A. centrale* obtained from Onderstepoort Biological Products (Pretoria, South Africa), *Anaplasma* sp. Omatjenne (in vitro culture material, obtained from E.P. Zweygarth, Freie Universität Berlin, Germany) and DNA from two jackal samples [31] confirmed to contain 16S rDNA sequences identical to *Anaplasma* sp. ZAM dog (deposited in GenBank under accession numbers MT918373 and MT918374). Note that the latter were initially described as *Anaplasma* sp. SA dog [31], but reanalysis of the sequence data after the *Anaplasma* sp. ZAM dog sequences became available in GenBank, confirmed that these sequences were identical to the *Anaplasma* sp. ZAM dog.

### 2.6. PacBio 16S rRNA Gene Sequencing

DNA samples (rodents = 25, dogs = 10, cattle = 9, and AFI patients = 9) previously tested using the qPCR assay were randomly selected for circular consensus sequencing (CCS) on the PacBio (Pacific Biosciences, Menlo Park, CA) platform to corroborate qPCR assay results. The 16S rRNA gene (V1-V8 variable regions) was amplified from the samples using barcoded universal 16S rRNA gene primers, 27F (5′-AGA GTT TGA TCM TGG CTC AGA ACG-3′) and 1435R (5′-CGA TTA CTA GCG ATT CCR RCT TCA-3′) [32,33] as previously described [34]. Sample-specific combinations of barcoded primers were used in a final reaction volume of 25 µL containing 1 X Phusion Flash^®^ High Fidelity PCR Master Mix (composed of Phusion Flash II DNA Polymerase, reaction buffer, dNTPs, and MgCl_2_; ThermoFisher Scientific, South Africa), 0.15 µM of each primer and 5 µL of template DNA (approximately 100 ng of DNA). For each sample, three technical replicates were performed using the same sample-specific barcoded primer set [34]. *Anaplasma centrale* vaccine strain (Onderstepoort Biological Products) was used as the positive control while PCR grade water was used as a no template negative control.

The thermal cycling parameters used were 98 °C for 30 s, followed by 35 cycles of 98 °C for 10 s, 60 °C for 30 s and 72 °C for 30 s, and a final extension at 72 °C for 10 min. PCR products were visualized by electrophoresis on a 1.5% agarose gel (1 × TAE buffer, pH 8.0) stained with ethidium bromide and viewed under UV light. PCR products were purified using the QIAquick^®^ PCR purification kit (Qiagen) according to the manufacturer’s instructions. Purified PCR products were subjected to CCS at the Genomics Sequencing Core of Washington State University, Pullman, USA. Appendix A shows the origin and list of samples used for CCS and multilocus gene sequencing.

### 2.7. Microbiome Sequence Data Analysis

Binning, trimming and filtering of the 16S rRNA amplicon sequence data was conducted using Pacific Biosciences software according to the set sequence size range and 99% precision. Reads were then analyzed using the Ribosomal Database Project (RDP) 16S classifier [35] to classify sequence reads to the genus level with a 95% confidence interval. Filtered data were then analyzed against the NCBI BLASTn 16S microbial database using the command line application to ascertain the identity of sequences. Sequence data were subsequently blasted against a local database created from *Anaplasma* spp. sequences downloaded from GenBank for precise assignment of *Anaplasma* spp. sequences within the microbiome data. Results from BLASTn were filtered to a minimum length of 1275 bp and 98% identity in Microsoft Excel [34]. Sequence reads that fell below 98% identity were reported at the genus level, while reads with an identity of 98% and above were reported at the species level [36,37,38]. Operational taxonomic units (OTUs) that were less than 1% of the total number of sequences were grouped as ‘rare’ [34].

### 2.8. Characterization of A. phagocytophilum by Multilocus Gene Sequencing

Four genes (16S rRNA, *gltA*, *msp4* and *ankA*) known to be useful for phylogenetic analysis of *A. phagocytophilum* [22,39,40,41,42] were amplified and sequenced from 32 qPCR-positive samples (cattle = 4; dogs = 11; AFI patients = 4; rodents = 8; ticks = 5 pools; Appendix A). Amplicons could not be obtained from the remaining qPCR-positive samples. Table 2 shows the primers used for amplification. For samples that did not produce amplicons with 16S and *gltA* primer set 1, a nested PCR was performed with primer set 2 as in Table 2.

Primers were used at a final concentration of 0.2 µM in a 20 µL reaction containing 10 µL of Phusion Flash^®^ High Fidelity PCR Master Mix (Thermofisher Scientific, South Africa) and 4 µL of template DNA. A second PCR using the same primers was performed using 2.5 µL of the primary PCR product as template. Cycling conditions were as recommended by the manufacturer, with 35 cycles for the primary PCR and 30 cycles for the second PCR. PCR products were purified using the QIAquick^®^ PCR purification kit (Qiagen, Germany), then cloned using the Clone Jet^®^ PCR Cloning Kit (Thermofisher Scientific, South Africa) according to the manufacturer’s instructions. Clones were screened by colony PCR and at least 10 positive clones per sample were sequenced on an ABI 3500XL Genetic Analyzer using vector primers pJET1.2F and pJET1.2R at Inqaba Biotechnical Industries (Pty) Ltd. (Pretoria, South Africa).

### 2.9. Sequence and Phylogenetic Analysis

16S rRNA, *gltA*, *msp4* and *ankA* gene sequences were assembled, edited and aligned using CLC Main Workbench 7.9 (Qiagen). Seven near full-length 16S rRNA *Anaplasma* spp. gene sequences obtained from the CCS datasets from three cattle, three dogs and a rodent were included in the 16S rDNA sequence alignment (C5, C13, C91, D24, D28, D36 and R98). Sequence identities were determined from GenBank using BLASTn [43]. Sequences were aligned with appropriate reference sequences from GenBank; sequence variation was inferred using the alignment tool in Workbench. Alignments were edited and trimmed in Bioedit 7 [44]. Bayesian inferences were deduced for 16S rRNA and *gltA* gene sequences using Mr Bayes 3.2 [45]. The best nucleotide substitution model predicted for the 16S rRNA gene sequences was generalized time reversible (GTR + I + G) using the Jmodel test 1.3 [46]. ProTest 3.0 predicted Jones-Taylor-Thornton (JTT + G) as the best model for GltA sequences [47]. Phylogenetic trees for the 16S rRNA and *gltA* genes were constructed using the maximum likelihood (ML) method carried out in PhyML 3.1 [48] and Bayesian inferences in Mr Bayes 3.2 [45]. Trees generated were edited in MEGA 7 [49]. Accession numbers of reference sequences are shown in Appendix A.

### 2.10. Data Availability

Sequences generated in this study were deposited under accession numbers MK814402-MK814450 and MK804077-MK804111 (Appendix A). Raw microbiome data from rodents, cattle, dogs and AFI patients is available at the sequence read archive (SRA) with BioProject accession numbers PRJNA546130 and PRJNA602191.

## 3. Results

### 3.1. Specificity of the qPCR Assay

Amplification of the *msp2* gene was not observed using control DNA from *A. marginale*, *A. centrale* and *Anaplasma* sp. Omatjenne; however, amplification occurred with *Anaplasma* sp. ZAM dog and *A. phagocytophilum* DNA. No amplification was observed from the negative control. Although we did not have *A. platys* control DNA to test, comparison of the qPCR primers and probes with a published *A. platys msp2* sequence [50] suggest that an amplicon would not be generated from *A. platys*, as there are eight and nine nucleotide differences in the forward and reverse primer target areas, respectively, and five nucleotide differences in the probe target sequence (Appendix A).

### 3.2. Anaplasma Phagocytophilum and/or Anaplasma sp. ZAM Dog DNA Occurred in All Hosts Tested

Based on the qPCR assay, 11% (8/74) of AFI patients, 59% (166/282) of wild rodents, 82% (46/56) of dogs, 85% (17/20) of *R. sanguineus* tick pools and 10% (10/100) of cattle samples were positive for *A. phagocytophilum* and/or *Anaplasma* sp. ZAM dog. The percentage of positive rodent samples from the different habitats ranged from 50 to 75% and was not statistically significantly different (chi-squared test, *p* = 0.24). For the *R. sanguineus* ticks, 80% and 90% of the pools from Athol and Hluvukani were positive, respectively. For cattle, 11% (2/19) of samples from Tlhavekisa, 5% (1/20) from Gottenburg, 15% (3/20) from Hlalakahle, 6% (1/18) from Utha, 13% (2/16) from Seville A and 14% (1/7) from Seville B were positive.

### 3.3. Sequence Analysis of Microbiome Data

PacBio CCS sequencing of 16S rRNA gene amplicons was conducted on samples from 10 dogs, 9 cattle, 25 rodents and 9 patients suffering from non-malarial AFI. Rarefaction curves were satisfied for all samples except from dog sample D28. Representative rarefaction curves are provided in Appendix A. Total numbers of sequence reads, mean number of reads per sample and salient findings are presented below for each host species.

#### 3.3.1. Dogs

PacBio CCS sequencing of 16S rRNA gene amplicons from DNA of 10 dog blood samples yielded 30,340 bacterial sequences. The mean number of sequences per sample was 3034 sequences. *Ehrlichia canis* made up 23.8% of the sequences, 19.3% of sequences were classified as *Anaplasma platys* and 14.8% as *Anaplasma* sp. ZAM dog, while 0.3% of the sequences from the blood corresponded to *A. phagocytophilum*. Other organisms detected were *Achromobacter xylosoxidans* (21.4% of the sequences), *Anaplasma* spp. (1.6%), ‘rare’ category (10.6%) and *Mycoplasma haemocanis* (4.9%) and *Achromobacter* sp. (2.1%). *Ehrlichia* spp. comprised of 1.1% of the total sequences from canine blood. Microbiome results from 40% of the dog samples corresponded to the results from the qPCR assay.

#### 3.3.2. Cattle

Sequencing of the 16S rRNA gene amplicons obtained from nine cattle samples resulted in 34,559 bacterial sequences. The mean number of sequences per sample was 3839 sequences. *Anaplasma marginale* made up 54% of the total bacterial sequences obtained from cattle blood followed by *Anaplasma* sp. Mymensingh with 22.2%, 10.5% of the sequences corresponded to *Anaplasma* spp. and 5.4% of sequences to *Anaplasma* sp. Dedessa. Sequence read prevalences were as follows: *Anaplasma* sp. Hadesa: 2.7%, *A. centrale*: 1.4%, *A. platys*: 0.2%, *Anaplasma* sp. Saso: 0.2% and *A. phagocytophilum*: 0.01% of the total sequences obtained from cattle. Three percent of the total sequences corresponded to the rare category. Other organisms of potential interest detected in far lower numbers include *Bartonella bovis* (0.4%), *Bartonella* spp. (0.03%) and *Ehrlichia minasensis* (0.02%). Microbiome results from 11.1% of the cattle samples corresponded to the qPCR assay, although all nine of the cattle samples tested positive using the qPCR. Microbiome analysis revealed that the nine cattle samples were infected with various combinations of the novel parasites *Anaplasma* sp. Mymensingh, *Anaplasma* sp. Dedessa, *Anaplasma* sp. Saso and *Anaplasma* sp. Hadesa. 16S rRNA sequences from *Anaplasma* sp. Mymensingh and *Anaplasma* sp. Dedessa are very closely related to *A. platys* and group closely with *A. phagocytophilum* and *Anasplama* sp. SA/ZAM dog. It is therefore possible that cross-reactions with these novel parasites also occur in the qPCR.

#### 3.3.3. Rodents

Microbiome sequencing of 25 rodent samples resulted in 65,060 bacterial sequences with a mean number of sequences per sample of 2602 sequences. *Bartonella grahamii* comprised of 29% of the total bacterial sequences in the rodents’ blood, while *Bartonella* sp. RF255YX comprised of 23%, and *Bartonella* spp. made up 12% of the sequences. Other organisms detected were *Pseudomonas* spp. (17.9%), *Ochrobactrum* spp. (7.2%), *Brucella* spp. (1%), *Anaplasma* spp. (0.5%), *B. henselae* (0.1%), *Ehrlichia* spp. (0.03%) and *Coxiella burnetii* (0.02%). The ‘rare’ group and unclassified OTUs made up 4.7% and 4.3%, respectively, of the bacterial sequences in the blood of the rodents. Ten sequences of *A. centrale*, five sequences of *A. phagocytophilum* and two sequences of *A. marginale* were detected from the rodents. Microbiome results from 60% of the rodents corresponded to the results from the qPCR assay.

#### 3.3.4. AFI Patients

Sequencing of 16S rRNA gene amplicons from nine AFI patients yielded 13,725 bacterial sequences. The mean number of sequences per sample was 1525 sequences. *Herbaspirillum huttiense* made up 27% of the total bacterial sequences obtained from the blood of the patients. Sequence read prevalences were as follows: *Rickettsia africae*: 16.1%, *Stenotrophomonas maltophilia*: 15.1% and *Stenotrophomonas* spp.: 11.3%, while 11.2% of the sequences were categorized in the ‘rare’ group. Other organisms detected in lower numbers include *Achromobacter xylosoxidans* (3.6%), *Delftia lacustris* (3%), *Sphingomonas paucimobilis* (2.8%), *Beijerinckia fluminensis* (1.9%), *Pseudomonas putida* (1.9%), *Rhizobium* spp. (1.9%) and *Sphingobium yanoikuyae* (1.2%), while *Rickettsia* spp. and *Herbaspirillum* spp. each comprised of 1% of the total sequences. Two sequences corresponding to *Brucella melitensis* were detected from a patient. No *Anaplasma* sequences were detected.

### 3.4. Multilocus Sequence Analysis of the 16S rRNA, gltA, msp4 and ankA Genes

The 16S rRNA, *gltA*, *msp4* and *ankA* gene sequences were analyzed from four AFI patients, eight rodents, four cattle, 11 dogs and five *R. sanguineus* tick pools. Based on 16S rRNA or *gltA* gene sequence analyses, none of the mammalian host species investigated were apparently infected with more than one *Anaplasma* species; except for dog D25, which was infected with *A. platys* and *A. phagocytophilum*. These data are described below and summarized in Table 3.

#### 3.4.1. 16S rRNA

Based on near full-length 16S rRNA gene sequences (1262–1465 nt), we identified two 16S rRNA sequence variants for *A. phagocytophilum* (Aph1/16S and Aph2/16S), two for *Anaplasma* sp. ZAM dog (Adog1/16S and Adog2/16S) and one each of *A. platys* (Apla1/16S), *Candidatus Anaplasma boleense* (Cab1/16S) and *Anaplasma* sp. Mymensingh (Asm1/16S; Table 3). The Aph1/16S and Aph2/16S sequences differed by 1 and 2 nt, respectively, from the *A. phagocytophilum* type strain Webster (U02521). Aph1/16S was obtained from one dog (D2; five identical cloned sequences), while Aph2/16S was obtained from one rodent (R98; *Mastomys natalensis* trapped in Hlalakahle) and two dogs (D24 and D28). A further three partial 16S rRNA gene sequences (690–693 nt) were obtained from two rodents (R102 and R103; *Rattus tanezumi* and *M. natalensis*, respectively; both trapped in Tlhavekisa) and one patient (H59) sample. While these sequences could be classified as *A. phagocytophilum* sequences, they could not be assigned to either Aph1/16S or Aph2/16S sequence variants due to the conserved nature of this region of the *A. phagocytophilum* 16S rRNA gene. This result highlights the importance of generating and using (near) full-length 16S rRNA gene sequences when assigning species designations and in constructing phylogenies.

The Adog2/16S sequences obtained in this study were identical to the *Anaplasma* sp. ZAM dog sequences previously described in Zambia (LC269823) [22], while the Adog1/16S sequences differed by a single deletion. Adog1/16S and Adog2/16S differed by 4 nt from the *Anaplasma* sp. SA dog sequences previously described in South Africa (AY570538, AY570539 and AY570540) [20]. Three near full-length Adog2/16S sequences were obtained from three dogs (D9, D27 and D36), while eight Adog1/16S sequences were obtained from four dogs (D3, D5, D27 and D37). A further eight partial sequences (628–1031 nt) obtained from two dogs (D36 and D37) were assigned as Adog1/16S, while eight partial sequences (687–698 nt) from four dogs (D3, D5, D27 and D37) and three *R. sanguineus* tick pools (RA3, RH3 and RH8) were assigned as Adog2/16S. Both Adog1/16S and Adog2/16S sequence variants were found in five dogs (D3, D5, D27, D36 and D37) suggesting that individual hosts may be infected with more than one *Anaplasma* sp. ZAM dog strain.

The nine *A. platys* 16S rRNA gene sequences obtained in this study from two dogs (D25 and D33) (Apla1/16S) were conserved and identical to the *A. platys* 16S rRNA gene sequences described from dogs in Zambia (LC269820, LC269821 and LC269822) [22] and to the corresponding 1383 nt of the 16S rRNA gene from the recently published genome of *A. platys* strain S3 [51]. Apla1/16S differed by 5 nt from the *A. platys* type strain 16S rRNA gene sequence (M82801) published in 1992 [52].

The *Candidatus Anaplasma boleense* sequence (Cab1/16S) obtained in this study from one cattle sample (C13; Seville A) differed by 1–2 nt from the *Candidatus Anaplasma boleense* sequences recently described from *Anopheles sinensis* from Wuhan, China (KU586025, KU586041 and KU586169) [53]. It differed by 3 nt from the *Candidatus Anaplasma boleense* sequences originally described from *Hyalomma asiaticum* ticks from the Bole region of Xinjiang, China (KJ410247, KJ410248 and KJ410249) [54] and by 3 nt from *Anaplasma* sp. Dedessa described from cattle from Southwestern Ethiopia (KY924886) [55].

The *Anaplasma* sp. Mymensingh sequences (Asm1/16S) obtained from two cattle samples (C5 and C91; from Hlalakahle and Seville A, respectively) were conserved and identical to *Anaplasma* sp. Mymensingh originally described from cattle in the Mymensingh district of Bangladesh (MF576175) [56].

#### 3.4.2. GltA

We identified one GltA sequence variant each for *A. phagocytophilum* (Aph1/GltA), and *Anaplasma* sp. ZAM dog (Adog1/GltA), based on partial GltA deduced amino acid sequences (138–302 aa; Table 3). *Anaplasma phagocytophilum* Aph1/GltA deduced amino acid sequences obtained from three dogs (D24, D25 and D28) and one rodent (R102) were conserved and identical to the sheep isolate *A. phagocytophilum* str. Norway variant2 (CP015376), and to *A. phagocytophilum* strain RD1 (SCV65315) and an *A. phagocytophilum* from *Ixodes* ticks (AKZ20811). Aph1/GltA differed by four amino acids from the *A. phagocytophilum* type strain Webster (AF304136).

One *Anaplasma* sp. ZAM dog GltA deduced amino acid sequence variant (Adog1/GltA) was identified in a dog (D36); it was identical to the *Anaplasma* sp. ZAM dog GltA sequence previously described in dogs in Zambia (LC269827) [22], and differed by one amino acid from *Anaplasma* sp. SA dog GltA sequences (AY570541 and AAT74599) [20].

#### 3.4.3. Msp4

Based on the deduced amino acid sequences (197–214 aa) obtained for Msp4, only one sequence type was identified (Table 3). Although the nucleotide sequence differed by 11 nt from the *A. phagocytophilum* type strain Webster (EU857674), the deduced amino acid sequence was identical to several published *A. phagocytophilum* Msp4 sequences.

A total of 23 Msp4 sequences were obtained from three patient samples (H27, H47 and H53), two dogs (D3 and D33), seven rodents from Tlhavekisa and Hlalakahle (R102, R103, R104 (*M. natalensis*), R105 (*Mastomys natalensis*), R124 (*Saccostomus campestris*), R125 (*Gerbillicus leucogaster*) and R138 (*Gerbillicus leucogaster*)) and from one cow sample (C42; Seville B). These were all conserved and identical at the amino acid level to *A. phagocytophilum* type strain Webster (ACH70064) and several other *A. phagocytophilum* Msp4 sequences described from various hosts (ACH70059, ACH70060, AGH02966, AGH02967, AGH02970, AGH02971, AHG97932, AJP32949, AVH68963 and AVH68960).

Alignments of *msp4* genes from several species (Appendix A) revealed that primer options were limited as the gene sequences were relatively variable between species, and the A + T content somewhat high (59% for *A. phagocytophilum*). Thus, it appears that the primers used in this study were not likely to anneal to *msp4* sequences from other known species and would only prime from the *A. phagocytophilum* gene. This hypothesis is supported by the fact that we failed to amplify the *msp4* gene from some samples that appeared to harbor only 16S rRNA gene sequences for a single novel *Anaplasma* species. For example, no *msp4* amplicons were obtained from samples D36, C5, C13 or C19; the only 16S rRNA sequence type detected in dog sample D36 was *Anaplasma* sp. ZAM dog, while cattle sample C13 corresponded only to *Candidatus Anaplasma boleense*, and cattle samples C5 and 91 corresponded to *Anaplasma* sp. Mymensingh.

Interestingly, one sample that appeared to harbor only *A. platys* (D33) 16S rRNA gene sequences was shown to contain an Msp4 sequence that was identical to *A. phagocytophilum* Msp4. With the recent publication of the *A. platys* genome [51] we see that the Msp4 sequences of *A. phagocytophilum* and *A. platys* are only 72% identical and are therefore distinguishable. This suggests that, although no *A. phagocytophilum* 16S rRNA or *gltA* gene sequences were detected in this sample, the sample was coinfected.

Several samples that appeared to harbor only *Anaplasma* sp. ZAM dog (D3, RA3, RH3 and RH8) 16S rRNA gene sequence variants were shown to contain an Msp4 sequence that was identical to *A. phagocytophilum* Msp4. Since the *Anaplasma* sp. ZAM dog Msp4 sequence is not known, it may be that it is identical to that of *A. phagocytophilum*, or that these samples were coinfected with *A. phagocytophilum*.

Although the Msp4 sequence that we detected matches known *A. phagocytophilum* sequences, we refrained from definitively assigning the sequence as being from *A. phagocytophilum* due to the fact that there is limited sequence data available from the more recently described *Anaplasma* spp., and the Msp4 sequence for *Candidatus Anaplasma boleense*, *Anaplasma* sp. Mymensingh and *Anaplasma* sp. ZAM dog, detected in our samples, is not known. Therefore, we referred to the sequence obtained in this study as *Anaplasma* sp./Msp4 (Asp/Msp4).

#### 3.4.4. AnkA

We amplified a short region of the *ankA* gene and based on the deduced amino acid sequences (137–144 aa), only one sequence variant was identified that was identical to *A. phagocytophilum* strain Dog2 AnkA (ADA72255), and to several other *A. phagocytophilum* AnkA sequences described from various hosts (AAS21270, ADV02358, ADV02361, ADV02363 and KJV67204). Seven AnkA sequences were obtained from three dogs (D24, D25 and D28), three rodents (R102, R103 and R124) and one *R. sanguineus* tick pool (RH1). Importantly, rodent R102 (*Rattus tanezumi*) had been shown to harbor *A. phagocytophilum* based on both 16S rRNA and *gltA* gene sequence analyses.

Dog sample D33 that appeared to harbor only *A. platys* using 16S rRNA gene sequence analysis, contained AnkA and Msp4 sequences that match those of *A. phagocytophilum*, and are clearly different from *A. platys*, suggesting that this sample was coinfected with *A. phagocytophilum*.

Comparison of the primers used to amplify *ankA* with *A. platys* and *A. marginale ankA* sequences indicate, respectively, 10 and 11 nt differences in the forward primer, and 13 and 8 nt differences in the reverse primer (Appendix A), suggesting that these primers would not prime over a broad range of *ankA* genes from different species.

The partial AnkA sequence obtained from dog D36 shown to harbor *Anaplasma* sp. ZAM dog by 16S rRNA and *gltA* gene sequence analyses is more confounding, and suggests that this sample may either be coinfected or that the AnkA sequence for *Anaplasma* sp. ZAM dog is the same as *A. phagocytophilum* in this region of the gene/protein.

As in the case of Msp4, there are no *Anaplasma* sp. ZAM dog, *A. bovis, Candidatus Anaplasma boleense* or *Anaplasma* sp. Mymensingh *ankA* gene sequences or deduced amino acid sequences available in the public databases. Although the detected sequence matches known *A. phagocytophilum* sequences, we refrained from definitively assigning the sequence as being from *A. phagocytophilum* due to this limited sequence availability. Therefore, we referred to the sequence obtained in this study as *Anaplasma* sp. AnkA (Asp/AnkA).

### 3.5. Phylogenetic Analyses

Phylogenetic trees were generated for the 16S rRNA gene and GltA peptide sequences. Sequence similarities observed in the 16S rRNA sequences were confirmed by phylogenetic analyses. The phylogenetic tree topologies obtained using three tree algorithms were similar, and the maximum likelihood tree was chosen as a representative tree (Figure 2). 16S rRNA phylogenetic analysis revealed similar topologies, with *Anaplasma* sp. SA dog and ZAM dog consistently grouping together in a separate clade from *A. phagocytophilum.*

Phylogenetic trees based on GltA peptide sequences were constructed (Figure 3); a phylogenetic tree generated from *gltA* gene sequences was similar to the GltA peptide-based tree (data not shown). Comparison with the 16S rRNA phylogenetic analysis revealed similar groupings, with *Anaplasma* sp. SA dog and ZAM dog consistently grouping together in a separate clade from *A. phagocytophilum.*

For both the Msp4 and AnkA amino acid datasets, the observed sequence similarities with *A. phagocytophilum* were confirmed by phylogenetic analyses (data not shown).

## 4. Discussion

This study provides the first report of the detection of the zoonotic agent *A. phagocytophilum* in humans, rodents, dogs and cattle in Mnisi, a rural community in South Africa. Analysis of the 16S rRNA gene PacBio circular consensus sequence data indicated the presence of low levels of *A. phagocytophilum* DNA in the blood of rodents, dogs and cattle in this study. This was confirmed by sequence analysis of 16S rRNA (humans, dogs and rodents) and *gltA* (dogs and rodents) genes. Reports of human granulocytic anaplasmosis occurring in Africa have been few [57,58]. In South Africa, there have been no reported *A. phagocytophilum* infections in humans. The significance of detecting *A. phagocytophilum* DNA in humans, dogs and rodents, and the potential role of *A. phagocytophilum* as a cause of AFI in South Africa, is not known.

Recent research in the Mnisi community [26] assessed the prevalence of selected zoonotic pathogens in patients that presented with non-malarial fever (≥37.5 °C) at the community health clinics. Organisms for which there was evidence of recent or past infection/exposure included *Bartonella* spp., spotted fever group *Rickettsia* spp., *Coxiella burnetii* and *Leptospira* spp., which could have been the cause of their fevers. Low levels of exposure to West Nile and Sindbis viruses, but not Rift Valley fever virus were found. No screening for *Anaplasma* species was done as part of that study [26].

In our study, we obtained one *A. phagocytophilum* (Aph) 16S rRNA sequence from patient H59. From another three patients, H27, H47 and H53, we obtained an Msp4 sequence that matched 100% to *A. phagocytophilum*, however, in an abundance of caution we refrained from definitively referring to this sequence as *A. phagocytophilum* Msp4 because we detected novel *Anaplasma* species in our study (not in the humans) for which Msp4 sequences have not yet been reported. These patients were found to be infected or exposed to other pathogens published in a separate study: IgG antibodies were detected to the spotted fever group *Rickettsia* spp. (H27, H47 and H59), *C. burnetii* (H47) and West Nile virus (H53); and IgM antibodies to Sindbis virus (H47) [26]. *Bartonella* spp. were detected by PCR in one patient (H27) [26]. In our study, *R. africae*, a spotted fever group rickettsia, was detected in the microbiome data from three patients (H18, H27 and H59). Although rarefaction curves were satisfied, *Anaplasma* spp. DNA was not detected in the microbiome data of the AFI patients tested, despite the fact that seven of them tested positive by qPCR, and targeted sequencing indicated that one was positive for the *A. phagocytophilum* 16S rRNA gene and three were positive for Msp4. Nevertheless, our data suggests that *A. phagocytophilum* was detected at low levels in samples from AFI patients. As these patients appear to be exposed to many pathogens, no causal link between a pathogen and AFI can be made, however, we suggest that the role of all of these pathogens should be considered in the investigation of febrile patients in rural areas of South Africa.

We detected *A. phagocytophilum* DNA in four dogs and three rodents (two *M. natalensis* and one *R. tanezumi* trapped in the urban/periurban area of Hlalakahle and in the communal rangelands of Tlhavekisa). This was based on 16S rRNA and GltA sequence analysis. It has been stated that domestic dogs play a role as sentinels of infection to humans [59]. *Mastomys natalensis* and *R. tanezumi* are also known synanthropes of humans; the close association of both species with humans indicates they are likely to serve as carriers of infection to man [60,61].

Only one sequence variant was found for both Msp4 and AnkA, even though this latter gene has been used in previous studies to show variability between *A. phagocytophilum* strains [62]. This may be because the region of the gene we targeted was short, unlike previous studies that amplified the entire open reading frame [63,64]. As we were using biobanked samples, or samples that were several years old, it was sometimes difficult to obtain full length gene sequences, particularly in the case of *ankA*, where the *A. phagocytophilum* gene is >3 kb in length. Nonetheless, it is important to note that although these sequences were identical to the *A. phagocytophilum* genes, we cannot be certain whether the sequences obtained from dogs and *R. sanguineus* were derived from *A. phagocytophilum* or *Anaplasma* sp. SA/ZAM dog because there are currently no *msp4* or *ankA* sequences available from this species for comparison. Note that in the phylogenetic analyses using 16S rRNA or GltA sequences, *Anaplasma* sp. SA/ZAM dog is quite closely positioned to *A. phagocytophilum*.

The partial Msp4 sequences were obtained from all host species investigated in this study, while the partial AnkA sequences were obtained from dogs, rodents and *R. sanguineus* ticks. Both Msp4 and AnkA sequences were found to be identical to *A. phagocytophilum.* Since no Msp4 or AnkA sequences from *Anaplasma* sp. ZAM dog, *A. bovis, Candidatus Anaplasma boleense* or *Anaplasma* sp. Mymensingh are currently available in the public databases we could not with any certainty assign the sequences obtained in this study to only *A. phagocytophilum*. Sequence alignments show that our primers would be unlikely to prime across clades (data not shown), that is not to say primers designed against *A. phagocytophilum* genes would be unlikely to prime against genes from *A. marginale*, *A. ovis*, *A. centrale* or *A. capra*. However, within the *A. phagocytophilum* clade, things will be more uncertain, and this includes *Anaplasma* sp. ZAM dog. More sequence data is needed to clarify this point.

In screening our sample set initially with a qPCR assay designed to specifically detect *A. phagocytophilum* DNA [29], we found that the test cross-reacted with *Anaplasma* sp. ZAM dog DNA, suggesting that the high number of qPCR positives obtained from dogs (82%) and *R. sanguineus* tick pools (85%) could probably be attributed to the presence of *Anaplasma* sp. ZAM dog. A large proportion of the 16S rRNA bacterial blood microbiome sequences from ten dog samples contained *Anaplasma* sp. ZAM dog sequences, confirming this supposition. An even larger proportion of the bacterial microbiome was classified as *A. platys*, but we did not have access to *A. platys* control DNA and therefore, although sequence alignments suggest that our qPCR test was unlikely to detect *A. platys*, we could not discount the possibility that the qPCR assay might also cross-react with *A. platys*. We subsequently identified *Anaplasma* sp. ZAM dog 16S rRNA gene sequence variants (Adog1 and Adog2) in six dogs and three *R. sanguineus* tick pools (Adog2). For one dog (D36), an *Anaplasma* sp. ZAM dog GltA sequence variant (Adog1) was also identified. In addition to this, *A. platys* was identified in a further two dogs based on 16S rRNA gene sequence analysis.

Not much is known about *Anaplasma* sp. ZAM dog and *Anaplasma* sp. SA dog apart from the initial studies reporting these sequences in dogs from South Africa and Zambia [20,22]. In South Africa, *Anaplasma* sp. SA dog was first described from three dogs presented at the Veterinary Teaching Hospital of the Medical University of South Africa [20]. It was shown to be genetically closely related to *A. phagocytophilum* based on 16S rRNA and *gltA* gene sequence analysis. Unfortunately, laboratory records and clinical data on these dogs were not available. In the subsequent Zambian study [22], similar 16S rRNA and *gltA* gene sequences were obtained from apparently healthy dogs and the organism was designated *Anaplasma* sp. ZAM dog. The authors also indicated dogs as a possible reservoir host in the transmission of this *Anaplasma* species. It should also be noted that these authors stated that “the same *Anaplasma* species” was previously reported in sheep in South Africa [65] and a goat in Mozambique [66]; and that transmission of this *Anaplasma* species was suspected to be through the bite of *R. sanguineus* ticks [20]. However, the species to which the authors were referring is in fact *Anaplasma* sp. Omatjenne, which groups phylogenetically with *A. platys*.

In sub-Saharan Africa, *A. platys* has been detected from dogs in North Central Nigeria [67], ticks from dogs in the Congo [68] and ticks and dogs from Cote d’Ivoire and Kenya [69]. In Southern Africa, it has been detected in *R. evertsi evertsi* collected from domestic and wild ruminants in South Africa [70], and more recently in domestic dogs in Zambia [22]. In Africa, *R. sanguineus* is thought to be the reservoir host that plays a role in the transmission of *A. platys* [68]; however, we did not detect *A. platys* in any of the *R. sanguineus* ticks sampled in this study.

The phylogenetic trees inferred from the 16S rRNA gene (Figure 2) and GltA (Figure 3) sequence data obtained in our study, consistently grouped *A. phagocytophilum*, *A. platys* and *Anaplasma* sp. SA/ZAM dog sequences into three distinct clades, indicative of a divergence between these organisms. The bootstrap values were, however, only poorly supportive of these relationships, which has been reported previously [20].

There is only a four-nucleotide difference between the 16S rRNA gene sequences of *Anaplasma* sp. ZAM dog and *Anaplasma* sp. SA dog; and only one amino acid difference in GltA. These data suggest that *Anaplasma* sp. ZAM dog and *Anaplasma* sp. SA dog are variants of the same species. Phylogenetic analyses furthermore grouped *Anaplasma* sp. ZAM dog and *Anaplasma* sp. SA dog into a distinct clade; providing sufficient divergence from other *Anaplasma* species to warrant classification as a separate species. Until appropriate type material can be deposited and the species can be formally described, we will refer to this novel organism as *Anaplasma* sp. SA dog for *Anaplasma* sp. Southern Africa dog. Our findings would suggest that *R. sanguineus* should be considered as a possible vector for *Anaplasma* sp. SA dog in South Africa. It will be necessary, however, to undertake a tick transmission study to confirm the vectorial capacity of this tick species.

We reported 16S rRNA gene sequences closely related to the novel organism *Candidatus Anaplasma boleense* from one heifer sample from the study area; this is also the first description of *Candidatus Anaplasma boleense* in South Africa. This agent was first detected from *Hyalomma asiaticum* collected from livestock in the Bole region of Xinjiang China. Phylogenetic analysis of 16S rRNA, *gltA* and *groEL* sequence data [54] revealed a lineage clearly differentiated from other *Anaplasma* species [54]. The organism was subsequently described from *Anopheles sinensis* mosquitoes in Wuhan, China [53]. The zoonotic potential and pathogenicity of this agent are unknown.

We also detected *Anaplasma* sp. Mymensingh 16S rRNA gene sequences from two cattle samples. *Anaplasma* sp. Mymensingh was originally described from cattle in the Mymensingh district of Bangladesh [56]. Phylogenetic analysis of combined 16S rRNA and *groEL* data [56] revealed that *Anaplasma* sp. (Mymensingh) clustered with *A. platys*. The zoonotic potential, pathogenicity, tick vector and reservoir hosts of this agent are unknown.

In conclusion, this study serves as the first report of the detection of *A. phagocytophilum* in humans, dogs and rodents in South Africa. We recommend that health care practitioners in the Mnisi community also consider *A. phagocytophilum* in the differential diagnosis of non-malarial AFI, which will help to guide appropriate treatment. The study furthermore provided evidence that *Anaplasma* sp. ZAM dog and *Anaplasma* sp. SA dog strain are phylogenetically distinct from other *Anaplasma* species and warrant classification as a separate species. We also report the first detection of *Candidatus Anaplasma boleense* and *Anaplasma* sp. Mymensingh in cattle in South Africa.

## Figures and Tables

**Figure 1 microorganisms-08-01812-f001:**
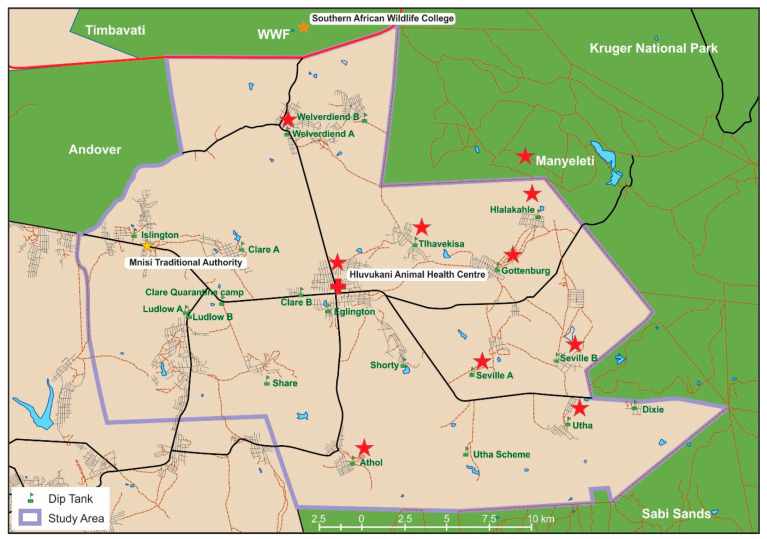
Map of the study area. The purple boundary shows the edge of the Mnisi community area. Red stars indicate locations of sample collections. Green indicates protected areas where wildlife roam freely.

**Figure 2 microorganisms-08-01812-f002:**
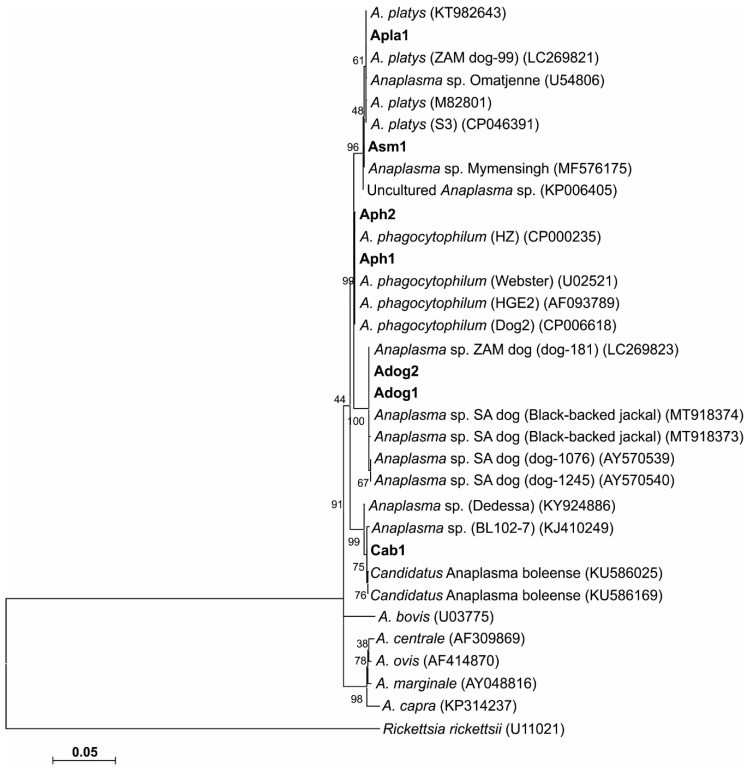
Maximum likelihood phylogenetic tree based on 16S rRNA nucleotide sequences in an alignment truncated to 1195 nt. The tree shows the phylogenetic relationship between the 16S rRNA gene sequence variants obtained in the study and related *Anaplasma* species. The 16S RNA variants were designated as *A. phagocytophilum* Aph1 and Aph2, *Anaplasma* sp. ZAM dog Adog1 and Adog2, *A. platys* Apla1, *Candidatus Anaplasma boleense* Cab1 and *Anaplasma* sp. Mymensingh Asm1. Sequence accession numbers are shown in parentheses. The evolutionary history was inferred by using the maximum likelihood method based on the general time reversible model. The tree with the highest log likelihood (−3042.29) is shown. The percentage of trees in which the associated taxa clustered together is shown next to the branches. The analysis involved 33 nucleotide sequences. All positions with less than 95% site coverage were eliminated. There was a total of 1186 positions in the final dataset. Evolutionary analyses were conducted in MEGA7 [49].

**Figure 3 microorganisms-08-01812-f003:**
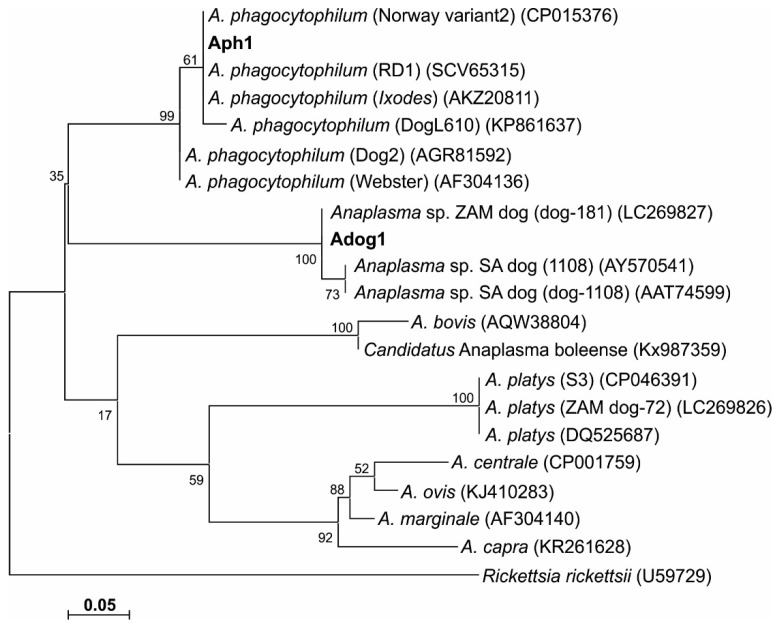
Maximum likelihood phylogenetic tree based on GltA deduced amino acid sequences showing the phylogenetic relationship between the obtained *Anaplasma* GltA sequence variants and related *Anaplasma* species. The GltA variants were designated as *A. phagocytophilum* Aph1 and *Anaplasma* sp. ZAM dog Adog1/GltA. Sequence accession numbers are shown in parentheses. The evolutionary history was inferred by using the maximum likelihood method based on the JTT matrix-based model. The tree with the highest log likelihood (−657.40) is shown. The percentage of trees in which the associated taxa clustered together is shown next to the branches. The tree is drawn to scale, with branch lengths measured in the number of substitutions per site. All positions with less than 95% site coverage were eliminated. There were a total of 52 positions in the final dataset. Evolutionary analyses were conducted in MEGA7 [49].

**Table 1 microorganisms-08-01812-t001:** Origin and sample sizes of the specimens used in the study.

Origin	AFI Patients ^†^	Rodents	Dogs ^‡^	Cattle ^‡^	Ticks *
**Protected area:**					
Manyeleti		76			
**Communal rangeland:**					
Tlhavekisa		35		19	
**Urban/periurban:**					
Athol					10 pools
Gottenburg	22	103		20	
Hlalakahle		63		20	
Hluvukani		5	56		10 pools
Seville A				16	
Seville B				7	
Utha	20			18	
Welverdiend	32				
Total	74	282	56	100	20 pools

* Ticks were male *Rhipicephalus sanguineus* collected from dogs. ^†^ Human blood samples were collected from acute febrile illness (AFI) patients that reported to the Gottenburg, Utha and Welverdiend clinics, but patients could have come from neighboring villages in the Mnisi community. ^‡^ The dog breed sampled was Africanis while blood samples from cattle were collected from Brahman (cross) and Sanga (typical) breeds.

**Table 2 microorganisms-08-01812-t002:** Primers used for amplification of four *Anaplasma* genes.

Gene	Primer Set	Primer Name	Primer Sequence (5′-3′)	Amplicon Length (bp/aa) *	Reference
16S rRNA	1	fD1	AGAGTTTGATCCTGGCTCAG	1470	[39]
		rP2	ACGGCTACCTTGTTACGACTT		
	2	16SAp-F ^†^	ATGGAGGATAATTAGTGGCAGA	700	This study
		16SAp-R ^†^	AAAAATCCCCACATTCAGCA		
*gltA*	1	F4B	CCGGGTTTTATGTCTACTGC	956/318	[41]
		1085R	ACTATACCKGAGTAAAAGTC		
	2	F1B ^†^	GATCATGARCARAATGCTTC	422/140	[22]
		1085R ^†^	ACTATACCKGAGTAAAAGTC		
*msp4*		AB1692F	TAATGATGCGTCTGATGTTAGCG	690/230	[42]
		AB1693R	CACCACCTGCTATGTTTACACG		
*ankA*		LA6-F	GAGAGATGCTTATGGTAAGAC	444/148	[40]
		LA1-R	CGTTCAGCCATCATTGTGAC		

* Amplicon length is in base pairs (bp), some analyses are converted to amino acids (aa). ^†^ For samples from which no amplicon could be generated using primer set 1, a second PCR, amplifying a shorter fragment of the gene, was attempted using primer set 2.

**Table 3 microorganisms-08-01812-t003:** *Anaplasma* species gene/protein variants detected in the Mnisi community.

Sample nr *	Origin		16S rRNA							GltA		Msp4	AnkA
	Location	Species	Aph ^†^		Adog ^‡^		Apla ^§^	Cab ^¶^	Asm ^#^	Aph	Adog	Asp **	Asp
			1 ^††^	2	1	2	1	1	1	1	1	M ^‡‡^	A ^§§^
**C5**	Hlalakahle	Cattle							X ***				
**C13**	Seville A	Cattle						X					
**C42**	Seville B	Cattle										X	
**C91**	Seville A	Cattle							X				
**D2**	Hluvukani	Dog	X										
**D3**	Hluvukani	Dog			X	X						X	X
**D5**	Hluvukani	Dog			X	X							
**D9**	Hluvukani	Dog				X							
**D24**	Hluvukani	Dog		X						X			
**D25**	Hluvukani	Dog					X			X			
**D27**	Hluvukani	Dog			X	X							
**D28**	Hluvukani	Dog		X						X			
**D33**	Hluvukani	Dog					X					X	X
**D36**	Hluvukani	Dog			X	X					X		X
**D37**	Hluvukani	Dog			X	X							
**H27**	Welverdiend	Human										X	
**H47**	Welverdiend	Human										X	
**H53**	Utah	Human										X	
**H59**	Utah	Human	U ^¶¶^										
**R98**	Hlalakahle	*M. natalensis*		X									
**R102**	Tlhavekisa	*R. tanezumi*	U							X		X	X
**R103**	Tlhavekisa	*M. natalensis*	U									X	X
**R104**	Tlhavekisa	*M. natalensis*										X	
**R105**	Tlhavekisa	*M. natalensis*										X	
**R124**	Hlalakahle	*Saccostomus* sp.										X	X
**R125**	Hlalakahle	*G. leucogaster*										X	
**R138**	Hlalakahle	*G. leucogaster*										X	
**RA1**	Athol	*R. sanguineus*										X	
**RA3**	Athol	*R. sanguineus*				X						X	
**RH1**	Hluvukani	*R. sanguineus*										X	X
**RH3**	Hluvukani	*R. sanguineus*				X						X	
**RH8**	Hluvukani	*R. sanguineus*				X						X	

* C, cattle; D, dog; H, human; R, rodent; RA, *R. sanguineus* (Athol); RH, *R. sanguineus* (Hluvukani); ^†^ Aph, *A. phagocytophilum*; ^‡^ Adog, *Anaplasma* sp. ZAM dog; ^§^ Apla, *A. platys*; ^¶^ Cab, *Candidatus Anaplasma boleense*; ^#^ Asm, *Anaplasma* sp. Mymensingh; ** Asp, *Anaplasma* sp.; ^††^ numbers represent sequence variants; ^‡‡^ M, Msp4 *Anaplasma* sequence; ^§§^ A, AnkA *Anaplasma* sequence; ^¶¶^ U, sequence variant unknown/could not be assigned due to short sequence length; *** X, sequence variant.

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
