# Peer review of "Anaplasma phagocytophilum and Other Anaplasma spp. in Various Hosts in the Mnisi Community, Mpumalanga Province, South Africa"

_microorganisms, 2020, doi:10.3390/microorganisms8111812_

Round 1

Reviewer 1 Report

Kolo, A. et al: Anaplasma phagocytophilum and other Anaplasma spp. in various hosts in the Mnisi community, Mpumalanga Province, South Africa.

This is a good, technically excellent study aimed at investigating „the importance of A. phagocytophilum and its role in febrile illness in South Africa“. Human cases of “non-malarial acute febrile illness (AFI)” and a collection of potential reservoir hosts/sentinels have been screened, and A.phagocytoplinum DNA indeed detected in some of them. In my view, however, the conclusiveness of this (otherwise nice) study is limited due to the authors’ omission to include also a representative group of AFI-free controls. The authors are, nevertheless, reasonably conservative in their conclusions as to the causal role of A.phagocytoplinum. Thus the most noteworthy output appears to be a ‘by-product’ – an identification of some more exotic Anaplasma spp. in various hosts inclusive of a possibly new Anaplasma sp. in dogs.

I’m convinced that the readership of Microorganisms will find this article interesting, and that it is worth publishing after only a minor revision.

Minor issues:

P.1, l.43: “ixodid hard ticks” is a tautology – either “ixodid-“  or “ hard ticks”..

P.3, l.103-7 / Table 1: a list of rodent species examined should be provided here or as a supplementary table.

P.3, l.109-11 / Table 1: the breeds of dogs and cattle could be specified (if known?).

P.8, l.400-1: which rodent species? Be specific, pls. (in order not to be needed to search for each item in supplementary tables..)

Table S1, row 27 and throughout: Utha

Table S2, row 33: Aedes albopictus

Table S2, row 39: Ixodes sp.

Table S2, row 40: “Boophilus” is an obsolete genus name, it has been synonymized with Rhipicephalus so the correct reference in this case is “Rhipicephalus microplus” (or “Rhipicephalus (Boophilus) microplus”, if the authors wish…).

Table S2, row 41: Haemaphysalis longicornis

Table S2, row 42: Ixodes sp.

Table S2, row 43: Ixodes sp.

Reviewer 2 Report

The study by Kolo et al. provides a thouroughful characterization of the presence and genetic diversity of Anaplasma phagocytophilum in different hosts of South Africa. This is an important contribution to the study of tick-borne pathogen epidemiology. The manuscript is well-written and logically organized. The introduction presents a balanced account of previous works on the topic and the discussion contrasts the results of the present study with those obtained by other authors. However, several points deserve the attention of the authors before considering the manuscript for publication.

Abstract

Line 25-26: Please reword ‘The test was found to detect…’

Methods

Line 91: Please, provide the coordinates of the sample collection sites. Also, specify what samples ‘were collected’. From the methodology described later in the paper, the authors only collected a set of the samples and the rest were obtained from sample banks.  

Line 105: It is not clear what the authors mean by ‘three research visits’. Field study? If yes, please, provide more details about it, what month(s) of each year? This information is not available in Table 1.

Line 106: Provide details about the ‘isoflurane’, what dose was used? What is the fabricant, and its city and country. How was the isoflurane administered to the animals?

Line 109: Specify what ‘Dog samples’ and ‘cattle samples’ were collected. How were the samples collected in these animals? Also provide information about the animals, e.g. age, breed, health status, origin (stray dogs?) and all relevant information that helps the reader understand the impact of the study.

Lines 111-112: Were the human samples collected for this study? If yes, provide information about consent of the patients to participate in the study. In the ‘Ethics statement’ above, the human samples are not mentioned. Please, clarify this point. If not, briefly explain the context in which the human samples were collected. The citation to reference ‘25’ is not enough.

Lines 114-115: The authors mentioned that the ticks were manually collected from dogs. However, it is not clear why if no dog was tested in Athol, tick samples were collected in this region. Why only male ticks were sampled?

Line 125: Before screening for pathogen presence, how the authors confirmed that the samples had in fact DNA of the quality needed for qPCR?

Lines 128-129: The sentence read as if only the ‘TaqMan probe’ was used for ‘identification’ However, primers are also involved in the ‘identification’. Please, reword.

Lines 134-135: Please, specify what is the ‘L610 [dog] strain’.

Lines 136-139: It is not clear the origin of A. marginale, A. centrale, Anaplasma sp. Omatjenne DNA. Please, explain and add the details in the methods.

Line 165: How the authors controlled the presence of contamination in their samples and/or during sample processing and sequencing ? Many tick species are incredibly difficult to surface sterile to the point of “environmentally clean”. Please see Salter’s 2014 paper (Reagent and laboratory contamination can critically impact sequence-based microbiome analyses).

What was the negative control used? Please, notice that that contaminant OTUs from sample laboratory processing steps can represent more than half the total sequence yield in sequencing runs (see Front Microbiol. 2020;11:1093.)

Did the authors test whether there was bias in the number of genera/species detected in a sample according to read numbers? Was there a positive correlation between the number of genera and the number of reads in individual samples? The authors mentioned something about this in results (lines 240-241), please, provide the rarefaction curves as supplementary material.

Did the authors sequence some host blood samples? If not, how can the authors know the contribution of host blood to the tick microbiota. Please, see Ticks Tick Borne Dis. 2018 Mar;9(3):563-572.

How the authors addressed the stability of their operational taxonomic units classification (see Microbiome. 2015; 3:20. doi: 10.1186/s40168-015-0081-x.) ?

Line 174: It is not clear why if the authors used ‘16S rRNA gene (V1-V8  variable  regions)’ that does not cover the full length of 16S rRNA gene then they filtered BLASTn results to a minimum length of 1275 bp and not to a minimum length equal to the length of the V1-V8  variable  regions of 16S rRNA gene.

Results

Lines 223-225: This is a decent solution for the problem of not having A.  platys DNA. However, this analysis should be done with an alignment of several available A.  platys msp2 sequences (as much as possible), and not using a single sequence. This is to consider the breadth of A.  platys msp2 sequence diversity in the region of the primers.

Lines 230-231: This is not informative enough. The authors should mention specifically how many samples tested positive for each of these pathogens (i.e. Anaplasma phagocytophilum and Anaplasma sp. ZAM dog). In addition, in the sections below (i.e. 3.3.1-3.3.4), the authors should mention explicitly on what proportion of the samples the results of the qPCR concurred with those of the next generation sequencing analysis. This is, were the 16S sequences of Anaplasma phagocytophilum and Anaplasma sp. ZAM dog found among the sequences identified by means of the next generation sequencing analysis? If 100% agreement was not found, please, discuss the reasons.

Line 238: Please, when describing the results do not use’reads’ and ‘sequences’ interchangeably. It is confusing because bacterial sequences result from analysis ‘reads’ and despite they are technically the same (i.e. base pairs), they refer to different stages of the analysis. Also, provide sequencing quality data, for example, as mentioned before the rarefaction curves, what proportion of the initial reads were discharged after quality checking.

Round 2

Reviewer 2 Report

The authors addressed all my previous concerns. I recommend publication.